# Automated—Mechanical Procedure Compared to Gentle Enzymatic Tissue Dissociation in Cell Function Studies

**DOI:** 10.3390/biom12050701

**Published:** 2022-05-14

**Authors:** Mariele Montanari, Sabrina Burattini, Caterina Ciacci, Patrizia Ambrogini, Silvia Carloni, Walter Balduini, Daniele Lopez, Giovanna Panza, Stefano Papa, Barbara Canonico

**Affiliations:** 1Department of Biomolecular Sciences, University of Urbino Carlo Bo, 61029 Urbino, Italy; mariele.montanari@uniurb.it (M.M.); sabrina.burattini@uniurb.it (S.B.); caterina.ciacci@uniurb.it (C.C.); patrizia.ambrogini@uniurb.it (P.A.); silvia.carloni@uniurb.it (S.C.); walter.balduini@uniurb.it (W.B.); d.lopez1@campus.uniurb.it (D.L.); g.panza1@campus.uniurb.it (G.P.); stefano.papa@uniurb.it (S.P.); 2Department of Pure and Applied Sciences (DiSPeA), University of Urbino Carlo Bo, 61029 Urbino, Italy

**Keywords:** tissue disaggregation, mechanical disaggregation (Medimachine™), enzymatic disaggregation, flow cytometry, confocal microscopy, transmission electron microscopy

## Abstract

The first step to obtain a cellular suspension from tissues is the disaggregation procedure. The cell suspension method has to provide a representative sample of the different cellular subpopulations and to maximize the number of viable functional cells. Here, we analyzed specific cell functions in cell suspensions from several rat tissues obtained by two different methods, automated–mechanical and enzymatic disaggregation. Flow cytometric, confocal, and ultrastructural (TEM) analyses were applied to the spleen, testis, liver and other tissues. Samples were treated by an enzymatic trypsin solution or processed by the Medimachine II (MMII). The automated–mechanical and enzymatic disaggregation procedures have shown to work similarly in some tissues, which displayed comparable amounts of apoptotic/necrotic cells. However, cells obtained by the enzyme-free Medimachine II protocols show a better preservation lysosome and mitochondria labeling, whereas the enzymatic gentle dissociation appears to constantly induce a lower amount of intracellular ROS; nevertheless, lightly increased ROS can be recognized as a complimentary signal to promote cell survival. Therefore, MMII represents a simple, fast, and standardized method for tissue processing, which allows to minimize bias arising from the operator’s ability. Our study points out technical issues to be adopted for specific organs and tissues to obtain functional cells.

## 1. Introduction

Tissues are highly complex ecosystems containing a diverse arrangement of cell subtypes. This has led to a rapid growth in studies attempting to capture cellular heterogeneity, and thereby gain a better understanding of tissue and organ development, normal function, and disease pathogenesis [1,2].

Tissue dissociation and its related problems have been described and defined over 110 years ago by Rous and Jones [3]. The first solid tumor disaggregation protocols generating viable cell suspensions were described in the late 1970s [4,5,6]; since then, various protocols have been elaborated [7,8,9]. These protocols are usually multistep procedures that involve one or a combination of mechanical, enzymatic, or chemical manipulations [8]. Enzymatic digestion commonly involves collagenase, hyaluronidase, dispase, trypsin, or DNase. Mechanical methods of tissue dissociation are based on aspiration, vortexing, scraping, or tissue pressing [10,11,12,13].

It is important to maximize the number of viable cells and prevent cell aggregates. However, in the suspensions prepared by these methods, some cell types can be selectively damaged [14]. Moreover, since manual mechanical methods are operator-dependent, the results are not highly reproducible. Several researchers have found that methods involving a combination of gentle mechanical action with enzymatic treatment have rendered better results in terms of yield and cell type representation, minimizing cellular aggregation [14,15]. Nevertheless, they are time-consuming and require a lot of handling [16].

The Medimachine System (CTSV) is an automated mechanical disaggregation system of solid human tissues without enzymes. The Medimachine spins the tissue in the Medicon filter (CTSV) to disaggregate tissue fragments into single cells. Below, we report some “historical” information on this instrumentation [17]. The use of the Medimachine II system is simple and intuitive: pop-ups depicting the various device options are visible on the touch screen and allow users to select whether to use the disaggregation or homogenization procedure, as described in the MM session.

The mechanical disaggregation from CTSV (and, from other Companies [18]) was previously adopted and compared to enzymatic digestion for some specific tissues [19].

There are several enzymatic approaches for tissue disaggregation, which have been optimized for cell culture or mammalian tissues [20,21,22]. Enzymatic digestion by trypsin and collagenase was first described by Rodbell (1964) [23]. It is a diffused method for the degradation of the collagen network of tissue. This method has some disadvantages such as the relatively high costs of enzymes, purity and purification of the enzymes, its time-consuming nature, and not always yielding reproducible results [24,25]. Deleterious effects of enzymes on the phenotype and behavior of cells [26,27] were demonstrated. However, very recently, different research groups have found excellent performances of collagenase (type I and II) in single viable cell obtainment from dental pulp [28] and from the atria and ventricles from the rat heart [29], with the addition of minimal concentrations of proteolytic enzymes. Indeed, cardiomyocytes, following a specific procedure, appeared viable and even cultivable [30], therefore preserving both proliferation and differentiation abilities.

Furthermore, Kakebeen and coworkers [22] in comparing multiple enzymatic approaches to disaggregate tissues from *Xenopus* embryos, did not find a significant difference in performance between liberase, trypsin, and papain, recommending researchers to use other starting tissues or with different target cells to compare all three enzymes for performance in their specific application.

We have given a rapid overview of the several options for performing tissue disaggregation; in fact the goal of the present study was not to indicate one best approach, but to provide information regarding the advantages and disadvantages of the protocols that we applied and their impact on the different managed tissues.

Indeed, since different groups adopted both the enzymatic and the mechanical approaches on some tissue, to improve cell yield and/or to minimize loss of viability, evaluations conducted in this study may be helpful to build the own “best performant procedure”, giving the suggestion of upstream-excluding one approach, due to any undesirable events occurring downstream.

The choice of enzymatic treatment with trypsin (a purely digestive enzyme with the ability to recognize different sites for the substrate) and the exclusion of treatment with collagenase can appear limiting, however, given the generic use of trypsin in the processes of extracting cells from tissues and organs, we chose to apply this gentle enzymatic disaggregation to be compared with the automated–mechanical one, by Medimachine II.

Tissue disaggregation remains one of the most important and problematic steps in solid tissue analysis by flow cytometry (FC). The method used for tissue disaggregation can have a definite impact on cell loss, viability, and different cell functions.

FC represents a powerful tool for understanding the heterogeneity of tumoral and normal tissues [31]. Along with the ability to interrogate millions of cells and characterize rare populations with respect to the expression of multiple parameters comes the loss of information on tissue location.

Traditionally, flow cytometry has been used to study the liquid tissues, blood, and bone marrow, and tissues that can easily be made into single cell suspensions, such as spleen and lymph node. However, with mechanical and enzymatic tissue dissociation, flow cytometry can be used to study normal and malignant cells isolated from solid tissue. The quality of the data depends on the careful preparation of samples, instrument calibration, and exclusion of sources of artifacts during data analysis.

In this study, by means of FC (for functional evaluations) and morphological ultrastructural analyses, we compared two frequently used methods of disaggregation, i.e., enzyme-free automated mechanical disaggregation (by Medimachine II) and low concentrations of the enzyme trypsin–EDTA (DNAse and BSA), to obtain a single cell suspension from different rat tissues, including organotypic slice cultures.

Flow cytometry experiments involving functional assays should be conducted shortly after the preparation of a single cell suspension or on cryopreserved live/thawed cells. The potential effects of cryopreservation should be tested in pilot experiments [21].

For these reasons, we also considered the possible impact of different disaggregation procedures on cryopreserved/thawed cells, in order to assess possible differences in post disaggregation freezing and thawing. These analyses are becoming increasingly important, as they would allow the most effective system to be developed to store biomaterial [32].

For instance, spermatogonial stem cells’ (SSCs) cryopreservation is an important method for the preservation of immature male fertility. However, freezing increases the production of intracellular reactive oxygen species (ROS) and causes oxidative damage to SSCs [30].

The functional cell response to the freezing and thawing processes may be influenced by the previous disaggregative procedures applied on the specific tissue, mainly the kind of disaggregation. Therefore, cell functions and ultrastructure were evaluated also on cryopreserved/thawed samples obtained from automated mechanical and enzymatic protocols.

## 2. Materials and Methods

### 2.1. Animals

Organs such as the spleen, testis, liver, and kidneys were harvested from freshly sacrificed Sprague-Dawley adult albino rats (Charles River, Milano, Italy) in accordance with the Italian law on animal experimentation (D.lgs. 26/2014; this research project was permitted with authorization N. 465/2015-PR by the Italian Ministry of Health). Spleen, testis, liver, and kidneys were uniformly minced with a scalpel to ~1 mm^3^ pieces. Minced tissue samples were then processed with the two disaggregation procedures.

In addition, 7 day-old Sprague-Dawley rats were also used (research project permitted with authorization N. 582/2020-PR by Italian Ministry of Health). From these animals, organotypic hippocampal slice cultures were prepared as previously described [33]. Briefly, hippocampi were dissected on ice and cut into 400 μm-thick transverse sections and plated onto Millicell culture inserts (0.4 μm Millicell-CM, Sigma Aldrich, St. Louis, MO, USA, Z354996-50EA). After preparation, hippocampal cultures were maintained for 2 weeks in a 37 °C humidified incubator gasified with a 5% CO_2_/95% O_2_ and with the culture medium being changed three times per week. Slices were then collected and processed with the two disaggregation procedures.

### 2.2. Cell Suspension Preparation

The Medimachine II instrument and Medicons disposable dissociators were supplied by CTSV s.r.l. (Turin, Italy). The tissues were dissected into Petri dishes containing cold PBS and then cut into 1 mm^3^ pieces.

For mechanical dissociation, 1 mm^3^ pieces of tissue were dissociated using the Medimachine System (CTSV s.r.l). In brief, disposable Medicon capsules that contain an immobile steel mesh with approximately 100 hexagonal holes, framed by six microblades, were filled with 1 mL of RPMI 1640 medium (Sigma Aldrich, St. Louis, MO, USA) before tissues were added. Medicons were then inserted into the Medimachine, and the machine was started. The Medimachine was run for a time ranging from 15 to 55 s at a constant speed of 100 rpm as described previously [34], resulting in an efficient cutting of the tissues. The cell suspension was aspirated with a syringe from the bottom part of the Medicon capsule. After the first run, 1 mL of medium was again added to the processed remaining tissue, and the cell fractions were subsequently pooled.

Enzyme cocktails were prepared with PBS solution with trypsin–EDTA 0.125%, BSA 5%, and DNAse 0.005%.

Briefly: for enzymatic dissociation, tissue fragments were treated by trypsin solution (containing trypsin–EDTA 0.125%, BSA 5%, DNAse 0.005%) for different time duration (10–30 min for different tissues) at 37 °C in a Petri dish, resuspended in medium with serum and then were mechanically disaggregated with a pipette or syringe needle gently.

Finally, both the cell suspension obtained from Medimachine and enzymatic procedure were filtered using a 70 μm Filcon (CTSV) for the brain, a 50 μm for liver, a 30 μm for spleen and kidney, and a 20 μm for testis. The steps of the two procedures and the final executed analyses are summarized in Figure 1, illustrating the different tissues processed in this study (spleen, kidney, liver, testis, cortical organotypic slices) and the applied technical approaches: flow cytometry, confocal microscopy and transmission electron microscopy.

### 2.3. Flow Cytometric Analyses on Fresh Samples

Cytometric experiments were performed with a FACSCanto II (BD) flow cytometer equipped with an argon laser (Blue, Ex 488 nm), a helium neon laser (Red, Ex 633 nm), and a solid-state diode laser (Violet, Ex 405 nm). The analyses were performed with the FACSDiva TM (BD) software. At least 10,000 cellular events were acquired for each sample.

Cell viability was assessed by means of propidium iodide (PI; Sigma-Aldrich) and 7-AAD staining. The cells were incubated for 10 min in the dark with PI 1 mg/mL and/or with 7-AAD (Beckman Coulter, Brea, CA, USA). We detected the percentage of PI- or 7AAD-positive events, obtaining a general index of cellular damage due to the different management of the samples.

Mitochondrial characteristics were investigated through TMRE staining. Tetramethylrodamine ethyl ester perchlorate (TMRE) (Sigma-Aldrich) is a cationic dye that is capable of penetrating the mitochondria and generating a red–orange fluorescence as intense as the mitochondrial membrane potential. TMRE 40 nM was added to the sample 15 min before the acquisition time. The samples were analyzed by flow cytometry using the appropriate fluorescence channel [35]. LysoTracker Green (LTG) dye (Thermo Fisher Scientific, Waltham, MA, USA) was used to mark and trace the lysosomes. The LysoTracker is an acidotropic and fluorescent probe which serves for the monitoring of acid organelles in living cells. The amount of fluorescence obtained by LysoTracker staining is directly proportional to the volume of lysosomes in the cell [36]. LysoTracker 100 nM was used to mark the lysosomes and after 30 min of incubation, the green lysosomal fluorescence was detected by flow cytometry and confocal microscopy [37].

The generation of reactive oxygen species was determined by the cytometric analysis of cells labeled with CM-H2DCFDA (Thermo Fisher Scientific, USA), which is capable of detecting the generation of intracellular H_2_O_2_. The 5- (e-6) -chloromethyl-2,7-dichlorodihydrofluorescein diacetate acetyl ester (CM-H2DCFDA) fluorescent probe is a membrane-permeable compound that is converted into a fluorescent, impermeable compound, H2DCF, by intracellular esterases. DCF (dichlorofluorescein) is a highly fluorescent compound produced by the oxidation of H2DCF by hydrogen peroxide. The amount of peroxide produced affects the intensity of DCF fluorescence inside the cells. CM-H2DCFDA was solubilized in DMSO (Sigma-Aldrich, Saint Louis, MO, USA) and then diluted in PBS and used at a final concentration of 5 μM for 30 min at 37 °C [38]. For each sample, at least 10,000 events were acquired thanks to the use of flow cytometry.

### 2.4. Flow Cytometric DNA Content Evaluation on Ethanol-Fixed Cells

Samples were fixed with cold 70% ethanol (−20 °C) and stored at 4 °C. At the time of analysis, samples were washed at least twice with PBS. The cell pellet was then resuspended in PBS containing propidium iodide (PI) 1 mg/mL and RNase 1 mg/mL. Finally, the sample was placed in a thermostatic bath at 37 °C for at least 30 min before acquiring events on the FACSCantoII cytometer.

### 2.5. Confocal Microscopy Analysis and Transmission Electron Microscopy (TEM)

Analyses were performed with a Leica TCS SP5 II confocal microscope (Leica Microsystem, Wetzlar, Germany) with 488, 543, and 633 nm lasers. Cell morphology was analyzed with the same instrument by means of phase contrast microscopy. The images obtained by the instrument were processed and analyzed by means of ImageJ software (NIH, UK).

For TEM investigations, the cells obtained from the two different disaggregations techniques were washed and fixed with 2.5% glutaraldehyde in 0.1 M phosphate-buffered saline for 1 h. The samples were post-fixed in 1% OsO_4_ for 1 h, dehydrated in alcohol, and incorporated into Araldite. Thin sections (60–80 nm) were stained with uranyl acetate and lead citrate and were analyzed using a Philips CM10 transmission electron microscope [39].

### 2.6. Statistical Analysis

Quantitative data are expressed as the mean ± SD based on at least three independent experiments. Analysis of variance (ANOVA) approaches were used to compare values among more than two different experimental groups for data that met the normality assumption. One-way ANOVA or two-way ANOVA were followed by a Bonferroni post-hoc test. A *p*-value < 0.05 was considered significant. All statistical analyses were performed using GraphPad Prism 5.0 (GraphPad software, San Diego, CA, USA).

## 3. Results and Discussion

### 3.1. Overall Cell Yield

The use of counting beads [40] not only allowed us to evaluate the absolute count of the events but also to compare the size of the cells with the size of the beads (Dako Cytocount: 5.2 μm and FlowCount beads: 10 μm). In this way, we could consider a size in micrometers for the events in the FSC vs SSC dot plot, although specific antibodies and/or markers are needed to distinguish all the cell types. The overall cell yield of the spleen, liver, testis, and kidney tissue are given in Appendix A, depicting the results from both the disaggregation procedures, and revealing higher cell numbers in the mechanical–automated disaggregation. Our results are only in partial agreement with the previous literature [34,41], however we adopted a gentle enzymatic protocol, without collagenase, which affects cell proliferation and the phenotypic pattern of tumor tissue-infiltrating immune cells (TTTII) [41]. This “weak” enzymatic disaggregation has shown to produce lower cell yields than the MMII procedures and with regard to more complex and “stronger” enzymatic protocols.

### 3.2. Spleen

The function of the spleen is to remove senescent red blood cells (RBCs) and circulating foreign material such as bacteria or cellular debris [42]. In hematology, the spleen serves as an important reservoir of monocytes [43,44], platelets [45,46], and memory B cells [47]. The spleen is also a significant site of hematopoiesis throughout vertebrate evolution and during fetal development in humans [48]. The mouse splenic tissue was disaggregated by the two methods (enzymatic and automated mechanics—Medi -) in order to obtain a comparison.

Furthermore, we processed both cryopreserved and fresh spleen samples, to highlight the morphological changes occurring during cryopreservation.

The experimental analysis carried out on fresh samples (Figure 2) allowed us to highlight two cell populations based on the size: big cells and small cells (Appendix A). Although our tests are mainly on cell functions and we did not evaluate antigenic profile, it is conceivable that the big cells are myeloid cells and small cells are lymphoid cells, in fact the frequencies of the cells are summarized in the Figure 1 and are consistent with both data from the literature and TEM analyses: in fact, besides APC, dendritic cells, macrophages, and monocytes [49], we found in marginal zone (MZ)-B cells in the rats that are slightly larger cells in terms of cell size [49].

Cells from the two methods of disaggregation were labeled by PI (Figure 2B) to evaluate cell surface damage. In Figure 2C,D, the morphological and ultrastructural characterization is shown, revealing well-preserved residual erythrocytes, with the characteristic biconcave disc morphology, in samples from the Medimachine II treatment. Nevertheless, although some erythrocytes still remain in the cell suspension, we focused the analyses on the main cell population represented by splenocytes, resulting in a mixed population of cells, as mentioned above, after both Medimachine and enzymatic disaggregation.

The lysosomal and mitochondrial behavior have been studied on both fresh (Figure 2M,N) and frozen samples, by TMRE and Lysotracker dyes, respectively. TMRE is a cationic dye that penetrates the mitochondria, generating a red–orange fluorescence as intense as the mitochondrial membrane potential. Lysotracker is a fluorescent acidotropic probe for the labeling and tracking of acidic organelles in live cells: it means that the amount of fluorescence obtained from the staining is directly related to the volume of lysosome-related organelles in a cell [36]. The statistical analysis shows a higher mean fluorescence intensity (MFI) of TMRE in samples obtained by the Medimachine II when compared to MFI values from Enzi. In particular, in the large cells (Figure 2N), a more extensive mitochondrial network was revealed, compared to the small cells, with higher TMRE MFI for the Medimachine II procedure. We also evaluated ROS content (Figure 2O), since mitochondrial ROS are implicated in the modulation of immune responses [50,51]. They are also involved in the regulation of B cell functions [52] and plasma cells are prone to produce large amounts of immunoglobulins causing the production of intracellular ROS [41], as macrophages and APC cells. The two procedures are equivalent for the detection of cell populations and the induction of cell damage, however they differ in lysosome and mitochondria network tracking; this was better revealed in cells obtained by Medimachine II, and, concomitantly, a ROS increase was detected.

#### 3.2.1. Cultural Assay of Differently Isolated Splenocytes

After obtaining the cell suspension following the two dissociation protocols and relative filtration, we separated the lymphocytes by density gradient centrifugation. The white ring of lymphocytes located between the two phases was collected and washed, then were treated with PHA and cultured for a week in RPMI, to stimulate proliferation [53].

We evaluated cell count and cell death in both control cultured cells, unstimulated and stimulated with PHA. On day one we found a slight increase in cell count in both experimental conditions, which was considered as being not relevant; after two days, PHA-treated samples showed an increase in count and low level of cell death (Figure 3A,B). On day three, we detected the highest cell count in samples obtained by Medimachine disaggregation, this contemporary finding the greatest PI-positive percentage. After 7 days, a strong reduction in the number of PHA-stimulated cells was evident, for splenocytes from both the Medi and Enzi procedures, reaching the low numbers of unstimulated residual cells, with a simultaneous massive increase in cell death in each condition.

Our data are in agreement with Autengruber and co-workers, finding that proteolytic enzymes can induce molecular alterations affecting immune cell functions such as proliferation [54].

#### 3.2.2. Cell Response of Differently Isolated Splenocytes to Cryopreservation and Thawing

Since in many cases the storage of the cells obtained by tissue dissociation is mandatory, to repeat the analyses or to postpone them, we evaluated the same functional parameters in the frozen–thawed isolated splenocytes, after 1 and 24 h of recovery. Data on cell surface damage (by PI uptake) (Figure 4A,B) appear strongly different from those obtained on fresh tissue, due to the Ficoll-gradient separation that we applied on each sample before the cryopreservation (Figure 2A), however data are similar for cells obtained by the two different procedures.

Staining by PI and LTG (Figure 4A) allowed us to obtain good cytometric profiles for cells from both enzymatic and Medimachine II procedures, with the significantly highest LTG MFI values for Medi-obtained small and big cells (Figure 4C), confirming the findings on fresh cells.

The same analyses were performed on thawed splenocytes recovered after 24 h in RPMI, maintained in a humidified incubator 37 °C, 5% CO_2_. Apoptotic behavior appeared here, strongly increasing the percentages of PI-positive events in a similar manner for Medi and Enzi samples (Figure 4D,E). Briefly, we can distinguish apoptotic cells [55] and necrotic cells in small cells (light purple events), whereas in big cells (green events) PI uptake exhibits a more brilliant fluorescence (+), suggesting a late phase of the apoptotic process and/or secondary necrosis: both the freeze and thaw processes impose considerable stresses on cells, which result in damage, metabolic dysfunction, and cell death [56]. These data point out that apoptotic enzymatic machinery is well preserved in cells by the two both procedures, since the percentages and distribution of PI positive events are similar.

We evaluated mitochondrial potential and ROS production (Figure 5A,B), finding evident differences between the two procedures for both ROS content and mitochondria membrane potential. Again, the highest ROS content was registered in cells obtained by the Medimachine II protocol, however the ROS increase does not imply a higher rate of PI-positive cells, as previously observed.

The disaggregation procedure should be gentle enough to maximize viability and minimize biologic alterations, yet robust enough to optimize cell yield and ensure that the final product accurately represents the in vivo cellular populations [57], therefore we can conclude that rat spleen disaggregation is efficiently performed by both procedures, as confirmed also by the flow cytometric DNA content evaluation (Figure 5C); these peculiar differences should be evaluated by the operator, and the selection of the procedure for the functional tests should take into account the cell response to disaggregation treatments.

### 3.3. Testicle

Our comparative data related to testicular tissues (TT) may provide useful suggestions for peculiar techniques such as in vitro spermatogenesis (IVS) [58,59]. To keep all options available, TT sampling is recommended, based on the literature.

We focused on the analysis of fresh tissue, evaluating the lysosomal content (by LTG), the mitochondrial network (by MTR) and the reactive oxygen species—ROS (by DCFDA) (Figure 6A,B). In fact, testicular and epididymal functions have been reported to be sensitive to the levels of ROS [50,51,52]. Therefore, maintaining a balanced redox state is critical for normal male reproductive functions [60].

The analyses carried out on rat testis allowed us to obtain two subpopulations of cells that were different in size: large and small cells. Taking into account the heterogeneity of the gonadal tissue and the morphologic/ultrastructural analysis, we can assume the large cells are the Sertoli cells and the still undifferentiated germ cells (Primary spermatocytes) and that the small cells are the spermatozoa and/or spermatids (Figure 6A).

The cytometric and morphologic analyses highlight as follows: large cells show a higher lysosomal content than small cells, revealing a higher content in lysosomes (Figure 6B). The Medi protocol demonstrates the better detection of lysosomes in small cells and significant lower percentages of PI+-damaged cells (Figure 6C). On the contrary, the enzymatic protocol highlights a brighter acid compartment (and fewer PI-positive events) in large cells. Mature spermatozoa exhibit the acrosome as a lysosome-like structure and the scenario drawn by our data is in agreement with the literature [61], which delineate interactions of acrosomal activity of sperm with several enzymes, including trypsin. Of note, the larger cells (Sertoli, Leydig, germ cells in the first steps of maturation) do not seem particularly sensitive to this enzymatic pretreatment.

In particular, the ultrastructure showed healthy Sertoli cells with an indentation in their cytoplasm enveloping the adjacent germinal elements (Figure 7). Prosecretory granules and mitochondria are scattered throughout the cytoplasm. [62]. Indeed, a greater content in lysosomes is consistent with the peculiarities of Sertoli cells. In fact, during spermatogenesis, most male germ cells undergo apoptosis, and the cytoplasmic portions of the elongating spermatids are shed as residual bodies, and are phagocytosed by Sertoli cells. After phagocytosis, they fuse with the abundant lysosomes and are recycled as energy sources for ATP production [63].

Mitochondria were also studied by flow cytometry and MTR staining: Figure 6C highlights significantly higher MTR MFI values in small cells (spermatozoo/spermatids) obtained by Medi disaggregation, whereas for big cells, the two procedures gave similar results. This better preservation of the mitochondrion (by mechanical automated protocol), cannot be understated, given the key role of this organelle for cellular homeostasis and sperm motility [64]. In fact, it is fundamental not only for sperm motility, but also for hyperactivation, capacitation, acrosome reaction, and fertilization [65].

Similarly, ROS were evaluated by the DCF dye (Figure 6D). The data obtained are in agreement with the literature [66], describing two proteins, semenogelin I and II, being abundantly expressed in the testicle, that may be altered by various enzymes such as trypsin. These proteins target the cell’s ROS-generating systems [67]. Microscopic analysis depicts a well-preserved sperm, for both protocols employed (Figure 6E,F), with a clear morphology of the rat sperm head and the midpiece, as detailed in Figure 6A, modified from [68]. Finally, histograms in Figure 6G show the DNA content of the Medimachine II and enzymatic-treated cells. In the first one, the different DNA amounts, typical of the different phases of the mitotic process, were well detectable, as described by Casuriaga and co-workers [16], on the contrary the same peaks were not observable for the cells obtained by the enzymatic treatment. Therefore, we concluded that the Medimachine II approach is optimally suitable for the testis disaggregation. Our data are in strong agreement with other groups [19,69,70].

#### Cell Response of Differently Isolated Testis Cells to Cryopreservation and Thawing

Analysis of lysosomal content was also carried out on the cryopreserved/thawed rat testicle cells (Figure 8A). The statistical data obtained confirmed that cryopreservation and thawing do not allow optimal results for cell function evaluations or evaluations of cell morphology, neither with the previous application of enzymatic disaggregation nor of mechanical dissociation, as the freezing process is the process that is responsible for the changes that take place in the structure and function of the cells. Nevertheless, Figure 8A–C highlights that the cells obtained by the Medimachine II disaggregation system highlight a better-detectable lysosomal network and, unexpectedly, a reduced ROS level, in both small and big cells.

Figure 8D,E shows the ultrastructure of the sperm tail, in fresh (Figure 8D,E) and frozen cells (Figure 8F,G). The microtubules are arranged in a central axoneme with two central single microtubules and nine peripheral double microtubules. Surrounding the microtubules are dense fibers, which provide rigidity and protection to the tail. Next is a layer of mitochondria which provides the energy for movement. Finally, the tail is wrapped in a protective outer sheath.

### 3.4. Liver

Rat liver was processed by both disaggregation procedures. The gating strategy in Figure 9A allowed for the exclusion of the most of debris to select the main cell populations. Bigger cells scattered in the FSC vs. SSC dot plots were dimensionally compared to counting beads of 10 microns: these appear as orange areas, whereas the red area represents the liver cell population and the blue represents the blood cells’ population. Propidium iodide staining at concentrations of 50 microgram/mL was applied to detect, as mentioned, not only the interruptions in the integrity of the plasma membrane, but also to distinguish necrotic and apoptotic cells, as observed in spleen samples (Figure 9E). These PI concentrations are able to highlight the nuclei also inside intact cells, as detected in Figure 9B by fluorescence microscopy, however, flow cytometric quantification can easily distinguish among dim, intermediate, and high fluorescent events, the latter representing the cells with membrane damage.

Figure 9E reveals lower percentages of damaged small (blood) cells and big hepatocytes after Medimachine II disaggregation. For liver sample management, before the analysis of lysosomal and mitochondrial functions and ROS levels, we report data on liver cell autofluorescence (Figure 9C,D). Autofluorescence (AF) emission of liver tissue depends on the presence of endogenous biomolecules that are able to fluoresce under suitable light excitation. Overall autofluorescence emission contains considerable information of diagnostic value because it is the sum of autofluorescence contributions from fluorochromes involved in metabolism, (i.e., NAD(P)H, flavins, lipofuscins, retinoids, porphyrins, bilirubin, and lipids) [71]. The interaction of biological tissues with electromagnetic radiation in the near-UV (≈300–400 nm), visible (≈400–700 nm), and near-IR (≈700–1100 nm) range gives rise to optical events, including light reflection, scattering, and absorption. We particularly consider cellular autofluorescence at different wavelengths: FITC (filter 530/30 DM 502 LP); PE (filter 585/42 DM 556 LP); PerCP-Cy5. (filter 670 LP; DM 655 LP), three ranges of emission from the 488 nm laser excitation, corresponding to the range of emission of the fluorescent probes employed in this study (Figure 9C). We have not compared healthy and pathologic liver tissues, however, we have analyzed autofluorescence and verified that enzymatic treatment increases the MFI values: this finding could be important because cellular AF varies with the cellular morphology, as well as with the metabolic and pathological states of cells, therefore, it can be used for diagnostic purposes [72] and should not be modified by dissociation/extraction procedures.

Lysosomes and mitochondrial behavior (Figure 9F,G) highlighted similar results for both disaggregation procedures, with the exception of the acidic compartment in small (blood) cells, which was better detected in the Medi-treated samples, and ROS content, which was higher in cells obtained by the same automated–mechanical protocol (Figure 9H). The confocal microscopy inset depicts a well preserved cell morphology and the fluorescent, massive mitochondrial network of the hepatocytes (Figure 9I).

To deeply characterize the morphology of the cells, since the precise identification of the cells, by means of mAb labeling, is lacking, we analyzed the samples by TEM. Ultrastructures are shown in Figure 10A (Medi-sample); the cell shows a well-preserved nucleus with euchromatin, heterochromatin, and nuclear pores within the nuclear membrane. A massive amount of mitochondria and peroxisomes are appreciable. In Figure 10B (Enzi sample) two hepatocytes are detectable: the first one appears with vacuolated cytoplasm, enriched in mitochondria, the second shows also some cisternae of the endoplasmic reticulum and golgi apparatus. The great amount of mitochondria, labeled by Mitotracker Green, is further underlined by the confocal image (Figure 10C). As mentioned, the disaggregation procedures led to a cell suspension with residual blood cells (Figure 10D). DNA content evaluation was efficiently performed in both the procedures (Figure 10E).

Since the liver is one selected organ on which we will continue the study, we employed different types of Medicons (with 50 μm steel blades and different polymer blades) (Appendix A).

These additional analyses suggest that a gentler mechanical dissociation could minimize the level of cell autofluorescence, collected particularly for FITC and PE. Furthermore, polymer blades demonstrated also a reduced impact on cell integrity and a weaker ROS induction (Appendix A).

#### Cell Response of Differently Isolated Liver Cells to Cryopreservation and Thawing

Cryopreservation remains the best option for the long-term storage of hepatocytes, providing a permanent and sufficient cell supply. However, isolated adult hepatocytes are poorly resistant to such a process, with a significant alteration both at the morphological and functional levels [73]. As expected, cytoplasmic vacuolization increased (Figure 11A,B) as a result of dehydration, which is typical of cryopreservation, although with the auxilium of DMSO and a considerable amount of FCS. Blood cells, as known, are less affected by the freezing and thawing (Figure 11C,D).

Figure 11E–N highlights results from surface cell damage (PI positivity), mitochondrial and lysosomal networks, together with ROS content, unexpectedly pointing out that Medimachine II-obtained cells show, in such preparations, a minor amount of ROS in respect of the mild enzymatic treatment. For the other parameters, the two both techniques seem almost equivalent, except for the lysosomal detection, once again emphasized in cells from mechanical automated protocols.

### 3.5. Other Tissues

#### Kidney and Organotypic Hippocampal Slice Cultures

Mouse kidney tissue was disaggregated by the two methods. This study represents an initial approach, since, as is this is often applied also for the liver, the kidney should be initially perfused, an approach that we have not applied in these preliminary kidney studies.

Briefly, the fresh rat cortical kidney sample was disrupted with Medicons 50 μm with a program of 40 s with a subsequent 10 s wash and filtered by a 30 μm Filcons filter, to discard aggregates and the largest debris.

Cell populations were selected, with a strategy that was able to exclude most of the debris. Figure 12A illustrates the distribution of cells obtained from kidney tissue, on FSC vs. SSC cytograms. We cannot define precisely which kidney cells they are, however in the kidney we know that endothelial, epithelial, mesangial, and podocyte cells coexist.

The ultrastructural analyses highlight a good morphology of an epithelial cell (Figure 12F), with typical microvilli (►), whereas in Figure 12F, the cell appears with cytoplasmic protrusion (→). Furthermore, a plethora of mitochondria, typical of various types of renal cells (epithelial of the proximal and distal tubule), are well detectable.

As usual, blood cells still remain in the final cell suspension (Figure 12L,M). Both enzymatic and automated mechanical protocols gave similar results for the TEM analysis and flow cytometric evaluation of cell functions. The percentage of PI-positive cells is low in both methods; however, it is even negligible in cells obtained by the enzymatic procedure (Figure 12B). Intriguingly, the data related to overall cell yields show that cell counts derived by the Medimachine II treatment are two-fold higher than those obtained by enzymatic dissociation. This finding suggests that the more damaged cells from the enzymatic method were definitely lost

Finally, the DNA content of cells obtained by the Medimachine II, shows CV (coefficient of variation) lower than the samples treated with enzymatic procedure (Figure 12N): this is able to simplify the possible application of this technical approach.

Results on cryopreserved/thawed kidney cells are shown in the Appendix A, showing Medimachine II-treated samples with cytoplasmic dehydration to a lesser extent than enzymatically obtained samples.

For starting evaluations on organotypic hippocampal slice cultures, we considered the overall condition of the samples and cell morphology. Both flow cytometry and confocal microscopy enabled us to distinguish between two subpopulations, named small and big cells, in the first phase of the investigations (Figure 13A–C).

To ensure that small and big cells’ distinction was not due to the chosen method of disaggregation, we compared the mechanical and enzymatic procedures for specific parameters such as cell size, morphology and TMRE and LTG fluorescence. Figure 13 highlights that both procedures guarantee similar results, though several researchers have published that automated-mechanical technical approach should be preferred due its efficiency to obtain single-cell suspensions from mouse brain tissue [74].

To verify if the differences observed in cell sizes could be correlated to different cellular lineage (i.e., neuronal or glial cells), we labeled cells with the astrocyte marker GFAP (Figure 13D).

## 4. Discussion

Tissues are complex mixtures of different cell subtypes, and this diversity is being increasingly characterized using high-throughput single cell analysis methods.

Enzymatic dissociation is typically carried out by introducing a digestion cocktail to minced, solid tissues and incubating them at specific temperatures, based on the enzyme cocktail being used. Enzyme strength and enzyme concentration are the two most important factors to consider when choosing a digestion cocktail for the preparation of a single cell suspension for flow cytometry. Lightly adherent cells should be isolated using a short digestion period with a gentle or mild enzyme to avoid these issues [21,75]. Determining the optimal strength and concentration of the enzymes being used in enzymatic dissociation is empirical and critical for the proper isolation of cells and successful digestion of tissues. By means of the Medimachine II, we applied an enzyme-free approach for mechanically dissociating tissue samples into viable single-cell suspensions. In fact, the success of dissociation procedures has been limited by their long processing times, low yield and high manual workload, leading to a significant increase in the overall analysis time [6]. Furthermore, enzymatic reactions can attack surface markers and cause alterations in gene expression profiles [54,74,76]. Tissue dissociation is therefore a careful balance between disrupting the tissue while not lysing the cells [77,78] These methods are now becoming progressively optimized [79,80,81] and are used both individually and in combination.

Described herein are protocols that use a gentle automated mechanical dissociation and a gentle enzymatic disaggregation with trypsin, DNAse, and EDTA, with the aim to isolate single cells from complex murine tissues.

A comparison between the two methods was performed for each tissue, for the functional parameters investigated. Cells obtained by the enzyme-free Medimachine II protocols showed the best lysosome and mitochondrial network labeling (by both FCM and confocal microscopy analyses). Particularly in spleen disaggregation, if lysosome–mitochondria functions and/or crosstalk are the main interest of the research, is preferable to adopt the mechanical procedure, taking into account a possible overproduction of ROS, whereas if we want to limit an additional, experimental induction of ROS, the enzymatic disaggregation is suitable.

Cell surface damage was equivalently induced by both procedures in splenocytes and in the largest kidney cells, whereas in testis disaggregation, the minor cell membrane damage was induced by the enzymatic protocol and, in liver dissociation, the best results were induced by the Medimachine II. Indeed, autofluorescence values investigated in murine hepatocytes were particularly elevated after the enzymatic treatment, hindering in this way, possible variations of autofluorescence emission, that are the sum of autofluorescence contributions from fluorochromes involved in metabolism [71].

The unique parameter in which the enzymatic gentle dissociation appears to have constantly better performance, with respect to the Medimachine treatment, is the lower ROS intracellular content. However, if this finding, on the one side, shows that the cells undergo a higher oxidative stress after mechanical automated disaggregation, on the other side, lightly increased ROS can be recognized as a complimentary signal to rescue the imbalanced ROS system, as well as to alarm stress and promote cell survival [82]. In fact, ROS systems are a new integrated network for sensing homeostasis and alarming stresses in organelle metabolic processes [82]. Furthermore, ROS are generated in almost every subcellular organelle in cells [83], including the plasma membrane [84], cytosol [85], mitochondria [86], nucleus [86], peroxisome [87], endoplasmic reticulum (ER) [88], Golgi [84], and others. For testicular tissue, ROS detection is particularly important. In the testes, they may be beneficial or even indispensable in the complex process of male germ cell proliferation and maturation, from diploid spermatogonia through meiosis to mature haploid spermatozoa [89]. Conversely, high doses, and/or inadequate removal of ROS caused by several mechanisms, can be very dangerous [63]. Studies indicate that low levels of ROS are crucial for spermatozoa to acquire their fertilizing ability [90,91,92].

As a final consideration, mechanical procedures were set for the “minimal manipulation” to avoid regulatory restrictions established by the Food and Drug Administration (FDA) in the US and worldwide, as previously stated by De Francesco and coworkers [19]. The FDA, particularly for the regenerative medicine field, considers enzymatic procedures to be “more than minimally manipulated” and have set heavy restrictions on them, while non-enzymatic methods are considered to be “minimally manipulated” by the FDA. In fact, although one limitation of our study is not having applied enzymatic cocktails containing collagenase, the use of strong enzymatic cocktails for the separation of cells from the extracellular matrix of tissue is considered a substantial manipulation, for specific applications [93].

Finally, another limitation of the work is the lack of a specific identification of the cells, these being uniquely divided into large and small.

During the continuation of the work, cell culture set up and proliferation studies will be further conducted on spleen and liver samples, on which we will distinguish cells on the basis of mAb labeling, where commercially available.

## Figures and Tables

**Figure 1 biomolecules-12-00701-f001:**
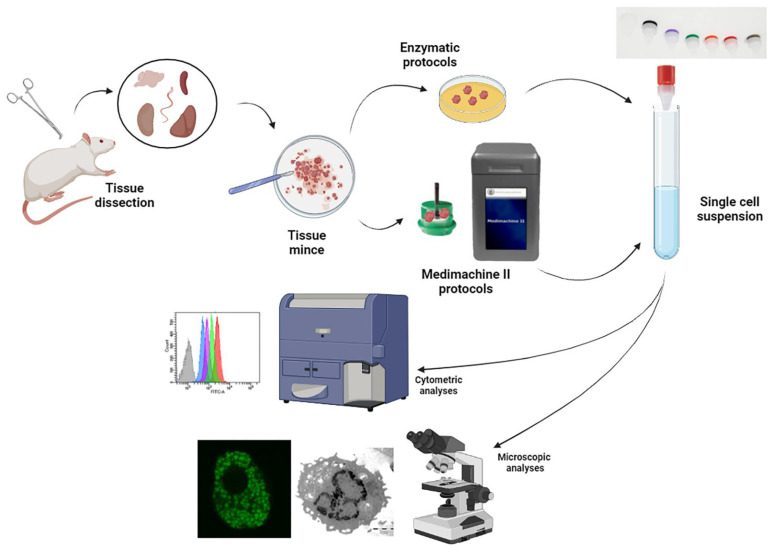
Scheme of the procedures applied for tissue disaggregation and for cell analyses (flow cytometry, confocal, and electron microscopy).

**Figure 2 biomolecules-12-00701-f002:**
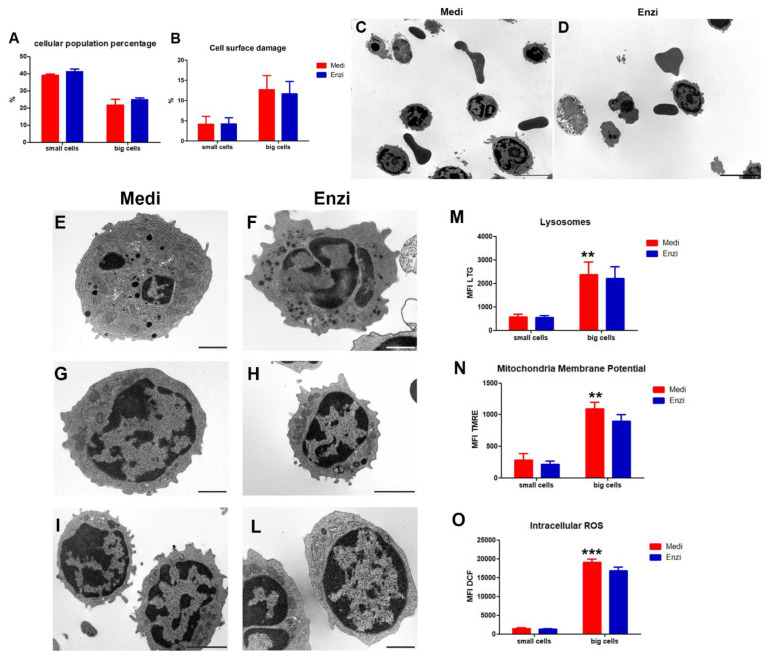
Mouse splenocytes were disaggregated by Medimachine II and an enzymatic approach. (**A**,**B**) Flow cytometric data of percentage cellular subpopulation and cell surface damage and ultrastructural observations (**C**,**L**). The Medimachine II treatment allowed for the acquisition of numerous splenocytes and well-preserved erythrocytes with the characteristic biconcave disc morphology. Plasma membrane and nuclear shape (**E**–**L**) of splenocytes at TEM show a good morphology. (**C**,**D**), Scale bars = 5 µm. Ultrastructural observations of the lymphocyte population after the Medimachine II (**E**,**G**,**I**) and enzymatic procedure (**F**,**H**,**L**). Morphology of the plasma membrane and nucleus are well-preserved. Mitochondria and RER present a good structural feature. (**E**–**G**,**L**), Scale bars = 1 µm; (**H**,**I**), Scale bars = 2 µm. (**M**) Statistical histogram of lysosomal network. Two-way ANOVA with a Bonferroni’s multiple comparison test revealed significant difference (** *p* < 0.01, *** *p* < 0.001) between the Medimachine II and enzymatic procedure. (**N**) Statistical histogram of mitochondrial membrane potential (Δψ-MMP). Two-way ANOVA with a Bonferroni’s multiple comparison test revealed a significant difference (** *p* < 0.01, *** *p* < 0.001) between Medimachine II and the enzymatic procedure. (**O**) Statistical histogram of intracellular ROS presence (H_2_O_2_ content). Two-way ANOVA with a Bonferroni’s multiple comparison test revealed a significant difference (** *p* < 0.01, *** *p* < 0.001) between Medimachine II and the enzymatic procedure. Protocol significance: Intracellular ROS ***; Lysosomes **; Mitochondria Membrane Potential **.

**Figure 3 biomolecules-12-00701-f003:**
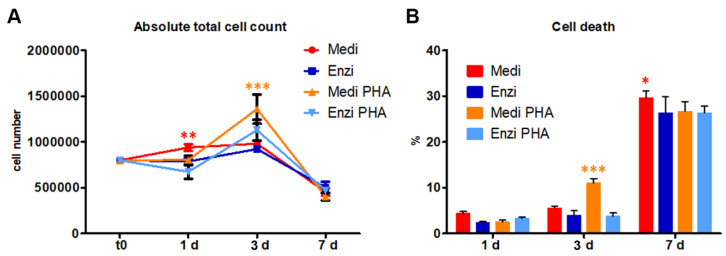
(**A**) Fold-increase in cell number, calculated from absolute for both procedures and the treatment with PHA at 1 d, 3 d and 7 d. (**B**) Statistical histogram of the percentage of cell death at 1, 3, and 7 d. Two-way ANOVA with a Bonferroni’s multiple comparison test revealed a significant difference (**p* < 0.05, ** *p* < 0.01, *** *p* < 0.001) between the Medimachine II/enzymatic procedure and the respective PHA-treated cells.

**Figure 4 biomolecules-12-00701-f004:**
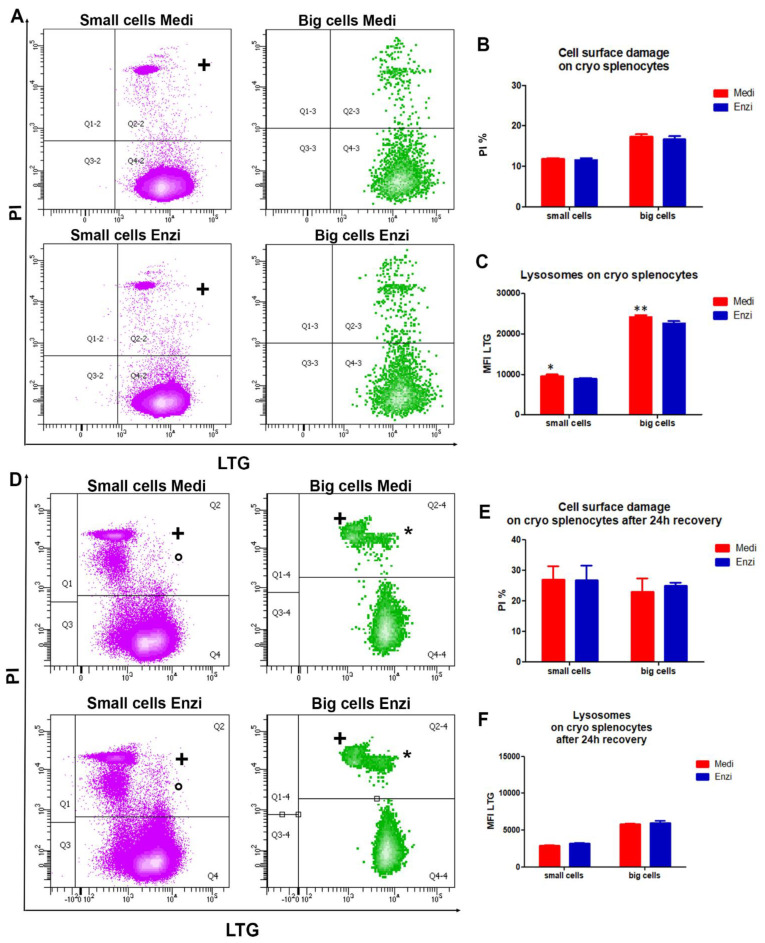
(**A**) Contour plot LTG versus PI on small cells and big cells for Medimachine II and enzymatic protocols of thawed/unfrozen splenocytes. (**B**) Statistical histogram of cell surface damage. (**C**) Statistical histogram of lysosomal network. Two-way ANOVA with a Bonferroni’s multiple comparison test revealed a significant difference (* *p* < 0.05, ** *p* < 0.01) between the Medimachine II and enzymatic procedure. (**D**) Contour plot LTG versus PI on small cells and big cells for Medimachine II and enzymatic protocols of thawed/unfrozen splenocytes after 24 h recovery. (**E**) Statistical histogram of cell surface damage percentage. (**F**) Statistical histogram of lysosomal network. Two-way ANOVA with a Bonferroni’s multiple comparison test revealed a significant difference (* *p* < 0.05, ** *p* < 0.01) between the Medimachine II and enzymatic procedure.

**Figure 5 biomolecules-12-00701-f005:**
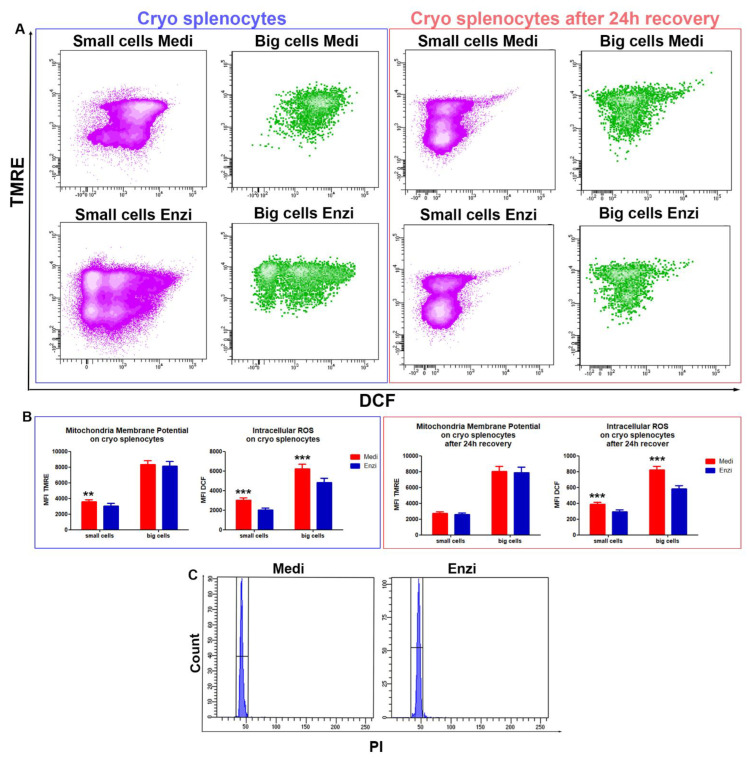
(**A**) Contour plot DCF versus TMRE on small cells and big cells for Medimachine II and enzymatic protocols of thawed/unfrozen splenocytes (Blue box above). Contour plot DCF versus TMRE on small cells and big cells for Medimachine II and enzymatic protocols of thawed/unfrozen splenocytes after 24 h recovery (Red box above). (**B**) Statistical histogram in blue box below of mitochondrial membrane potential (Δψ-MMP) and intracellular ROS presence (H_2_O_2_ content) of thawed/unfrozen splenocytes. Statistical histogram in red box below of mitochondrial membrane potential (Δψ-MMP) and intracellular ROS presence (H_2_O_2_ content) of thawed/unfrozen splenocytes after 24 h recovery. Two-way ANOVA with a Bonferroni’s multiple comparison test revealed a significant difference (** *p* < 0.01, *** *p* < 0.001) between the Medimachine II and enzymatic procedures. Under flow cytometry histograms of cell cycle analyses of both procedures. (**C**) Flow cytometry histograms of cell cycle analyses of both procedures of fresh spleen cells.

**Figure 6 biomolecules-12-00701-f006:**
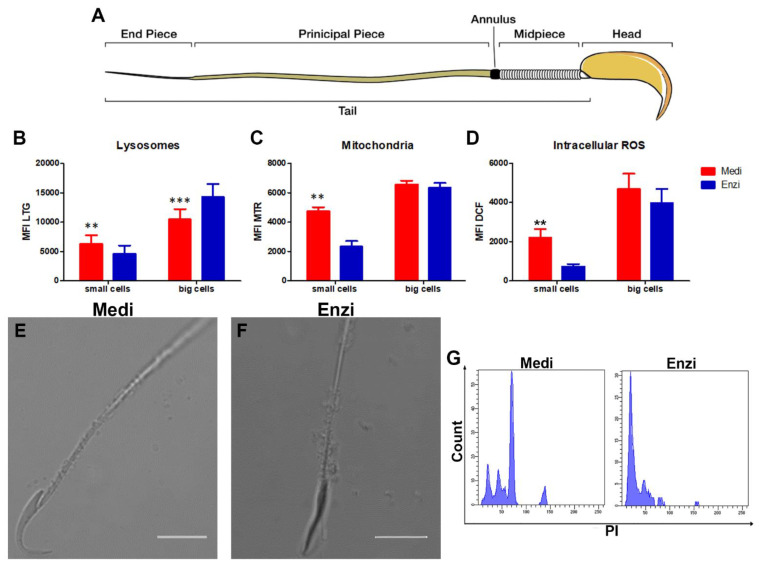
(**A**) A cartoon of a mature sperm of a rat. (**B**) Statistical histogram of the lysosomal network. Two-way ANOVA with a Bonferroni’s multiple comparison test revealed a significant difference (** *p* < 0.01, *** *p* < 0.001) between the Medimachine II and enzymatic procedure. (**C**) Statistical histogram of mitochondrial compartment. Two-way ANOVA with a Bonferroni’s multiple comparison test revealed a significant difference (** *p* < 0.01, *** *p* < 0.001) between the Medimachine II and enzymatic procedure. (**D**) Statistical histogram of intracellular ROS presence (H_2_O_2_ content). Two-way ANOVA with a Bonferroni’s multiple comparison test revealed a significant difference (** *p* < 0.01, *** *p* < 0.001) between the Medimachine II and enzymatic procedure. Protocol significance: Lysosomes **; Intracellular ROS **; Mitochondria Membrane Potential ***. (**E**,**F**) The analysis under a confocal microscope made it possible to highlight the sperm head, with the typical hook shape, allowing us to appreciate the preserved integrity of the tail, both with the enzymatic and mechanical dissociation procedure. (**G**) Flow cytometry histograms of cell cycle analyses of cells from both procedures.

**Figure 7 biomolecules-12-00701-f007:**
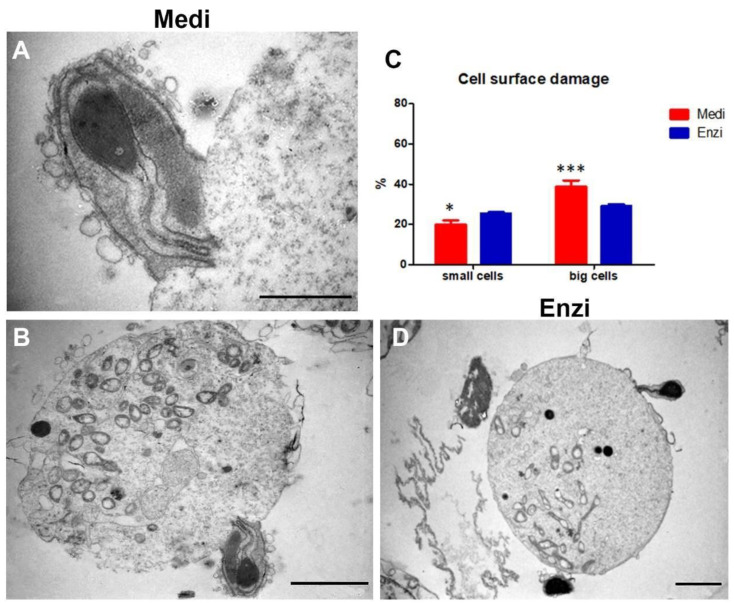
Ultrastructural study of the cells from the testicle after Medimachine II (**A**,**B**) and enzymatic procedures (**D**). In both methods, one can observe the close relationship between spermatozoa and/or spermatids and spermatogonia and/or Sertoli cells. Sperm heads showing a well-shaped nucleus with condensed chromatin are visible. The tail is recognizable. (**A**), Scale bar: 500 nm; (**B**,**D**), Scale bars: 2 µm. (**C**) Statistical histogram of cell surface damage percentage. Two-way ANOVA with Bonferroni’s multiple comparison test revealed significant difference (* *p* < 0.05, *** *p* < 0.001) between Medimachine II and enzymatic procedure.

**Figure 8 biomolecules-12-00701-f008:**
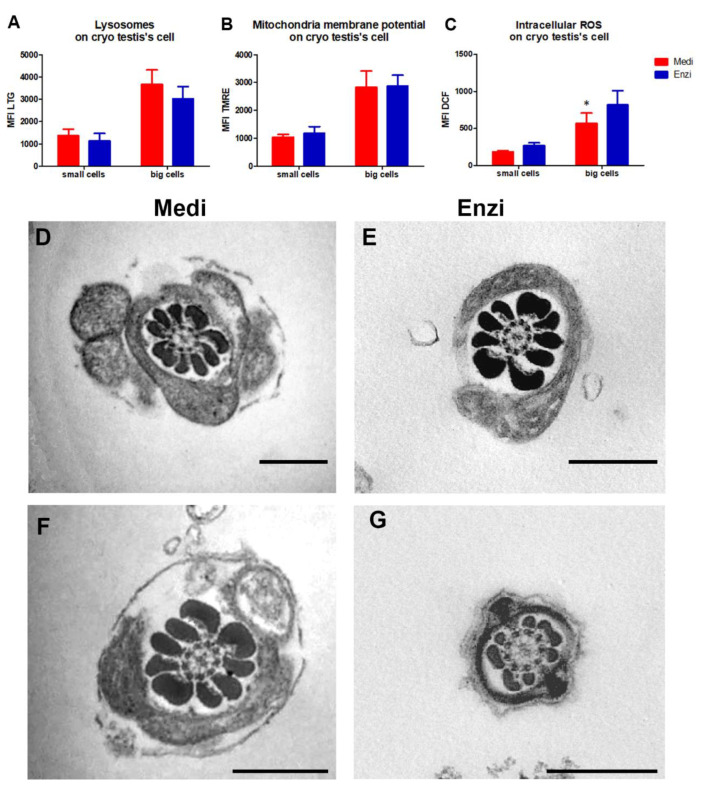
(**A**) Statistical histogram of lysosomal network. Two-way ANOVA with a Bonferroni’s multiple comparison test revealed a significant difference (* *p* < 0.05,) between the Medimachine II and enzymatic procedure. (**B**) Statistical histogram of mitochondrial compartment. Two-way ANOVA with a Bonferroni’s multiple comparison test revealed a significant difference (* *p* < 0.05) between the Medimachine II and enzymatic procedure. (**C**) Statistical histogram of intracellular ROS presence (H_2_O_2_ content). Two-way ANOVA with a Bonferroni’s multiple comparison test revealed a significant difference (*p* < 0.05) between the Medimachine II and enzymatic procedure. Protocol significance: Lysosomes *; Intracellular ROS *. (**D**,**E**) TEM micrographs of fresh samples showing a cross-section of a mouse sperm tail with normal axonemal pattern (showing the regular microtubule organization with nine microtubule doublets surrounding the central pair (9  +  2)). (**F**,**G**) TEM micrographs of cryo-sample showing a cross-section of a mouse sperm tail with normal axonemal pattern. Scale bars: 500 nm.

**Figure 9 biomolecules-12-00701-f009:**
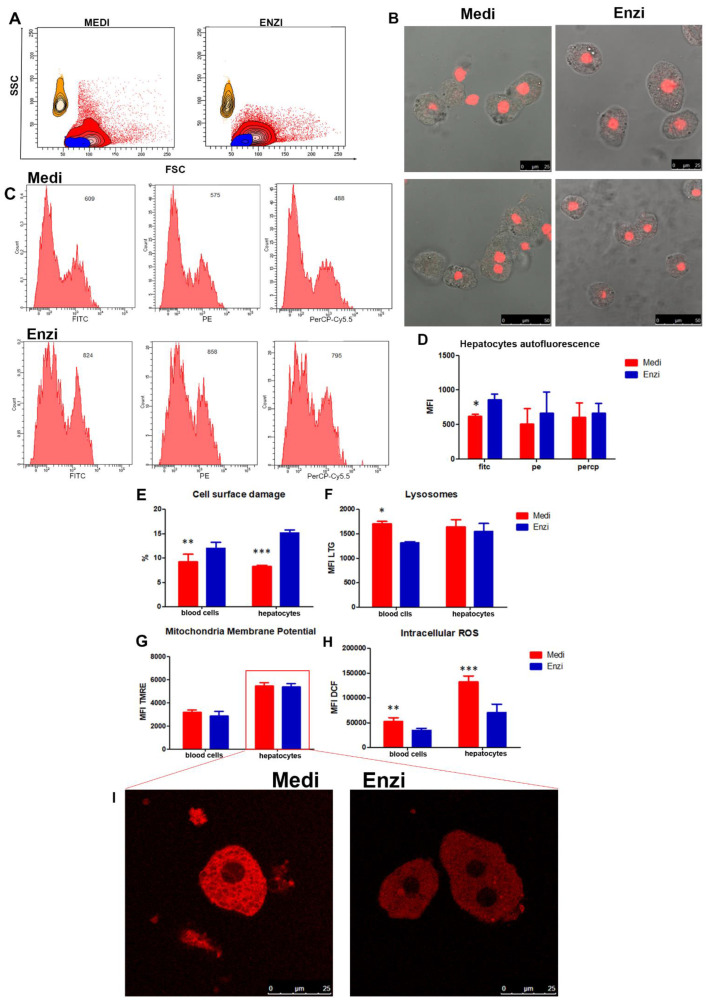
(**A**) Gating cell populations of blood cells (small cells, Blue), hepatocytes (big cells, Red) in the SSC versus FSC contour plot, distinguished by beads’ size (beads, Orange) for the further analysis. (**B**) Representative microscopic pictures (bright field) of hepatocytes obtained from both procedures, the nucleus (red) was stained with propidium iodide. (**C**) Representative cytometric histograms of hepatocytes autofluorescence for both procedures in Fitc, PE, Percp. (**D**) Statistical histogram of hepatocytes related autofluorescence for both procedures in Fitc, PE, Percp. (**E**) Statistical histogram of cell surface damage percentage. Two-way ANOVA with a Bonferroni’s multiple comparison test revealed a significant difference (* *p* < 0.05, ** *p* < 0.01, *** *p* < 0.001) between the Medimachine II and enzymatic procedure. (**F**) Statistical histogram of lysosomal network. Two-way ANOVA with a Bonferroni’s multiple comparison test revealed a significant difference (* *p* < 0.05, ** *p* < 0.01, *** *p* < 0.001) between the Medimachine II and enzymatic procedure. (**G**) Statistical histogram of mitochondrial membrane potential (Δψ-MMP). Two-way ANOVA with a Bonferroni’s multiple comparison test revealed a significant difference (* *p* < 0.05, ** *p* < 0.01, *** *p* < 0.001) between the Medimachine II and enzymatic procedure. (**H**) Statistical histogram of intracellular ROS presence (H_2_O_2_ content). Two-way ANOVA with a Bonferroni’s multiple comparison test revealed a significant difference (*p* < 0.05, ** *p* < 0.01, *** *p* < 0.001) between the Medimachine II and enzymatic procedure. Protocol significance: Lysosomes *; Intracellular ROS ***. (**I**) Representative confocal imaging of the TMRE-related fluorescence of hepatocytes obtained by the Medimachine II and enzymatic methods.

**Figure 10 biomolecules-12-00701-f010:**
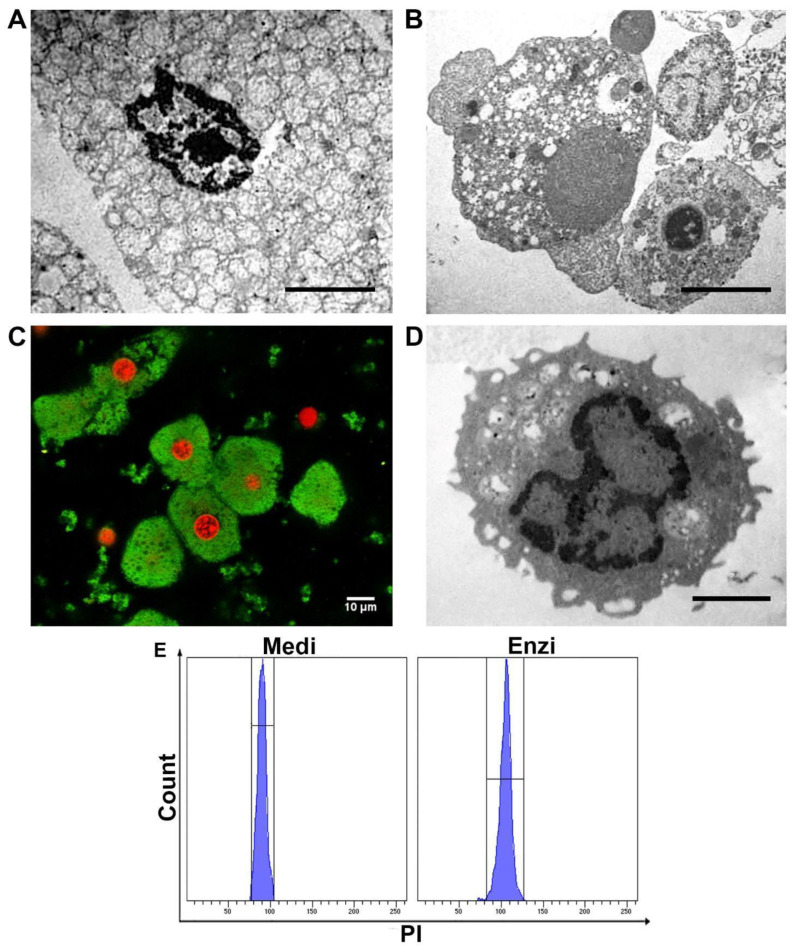
(**A**) Transmission electron microscope section of a hepatocyte obtained from disaggregation by Medimachine II. (**B**) Transmission electron microscope section of a hepatocyte obtained from enzymatic dissociation. (**C**) Confocal section of liver cells obtained from disaggregation with Medimachine II, in green MTG-labeled cytoplasm and in red PI-labeled nuclei. Scale bar: 10 μm. (**D**) Transmission electron microscope section of a lymphocyte obtained from enzymatic procedure. (**A**,**B**,**D**). Scale bars: 2 μm. (**E**) Flow cytometry histograms of cell cycle analyses of both procedures.

**Figure 11 biomolecules-12-00701-f011:**
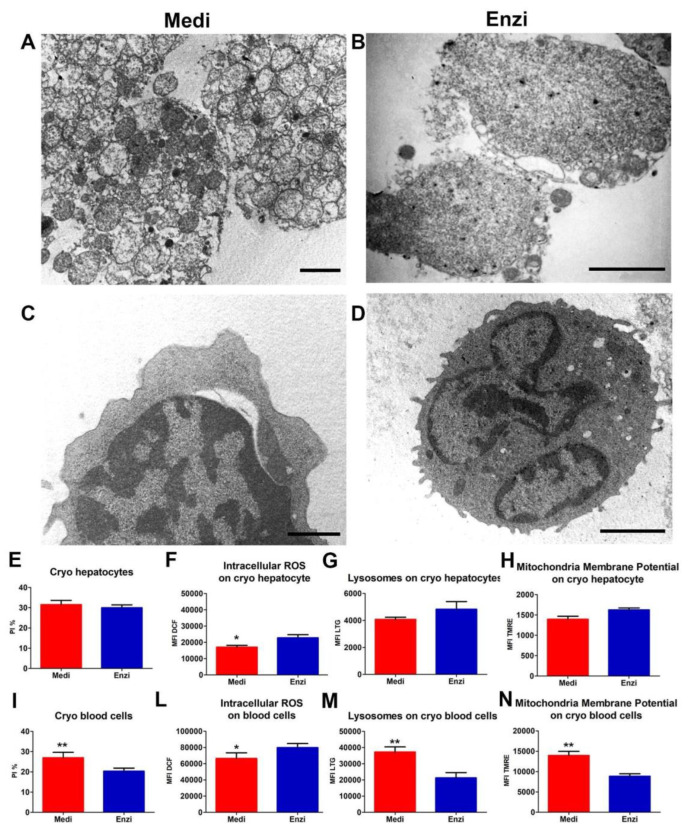
Ultrastructural study of the cryo cells from the liver. (**A**,**B**) TEM of an ultrathin section of hepatocytes with nucleus and many mitochondria in both procedures; Scale bars: 2 µm. (**C**,**D**) TEM of an ultrathin section of blood cells (lymphocytes) with nucleus and many mitochondria. Scale bars: 1 µm. (**E**) Statistical histogram of cell surface damage percentage of hepatocytes. Paired t-test revealed a significant difference (* *p* < 0.05, ** *p* < 0.01) between the Medimachine II and enzymatic procedure. (**F**) Statistical histogram of intracellular ROS presence (H_2_O_2_ content) of hepatocytes. Paired t-test revealed a significant difference (* *p* < 0.05, ** *p* < 0.01) between the Medimachine II and enzymatic procedure. (**G**) Statistical histogram of lysosomal network of hepatocytes. Paired t-test revealed a significant difference (* *p* < 0.05, ** *p* < 0.01) between the Medimachine II and enzymatic procedure. (**H**) Statistical histogram of mitochondrial membrane potential (Δψ-MMP). Paired t-test revealed a significant difference (* *p* < 0.05, ** *p* < 0.01) between the Medimachine II and enzymatic procedure. (**I**) Statistical histogram of cell surface damage percentage of blood cells. Paired t-test revealed a significant difference (* *p* < 0.05, ** *p* < 0.01) between the Medimachine II and enzymatic procedure. (**L**) Statistical histogram of intracellular ROS (H_2_O_2_ content) in blood cells. Paired t-test revealed a significant difference (* *p* < 0.05, ** *p* < 0.01) between the Medimachine II and enzymatic procedure. (**M**) Statistical histogram of lysosomal network of blood cells. Paired t-test revealed a significant difference (* *p* < 0.05, ** *p* < 0.01) between the Medimachine II and enzymatic procedure. (**N**) Statistical histogram of mitochondrial membrane potential (Δψ-MMP) in blood cells. Paired t-test revealed a significant difference (* *p* < 0.05, ** *p* < 0.01) between the Medimachine II and enzymatic procedure. (**G**) Statistical histogram of mitochondrial membrane potential (Δψ-MMP). Two-way ANOVA with a Bonferroni’s multiple comparison test revealed a significant difference (* *p* < 0.05, ** *p* < 0.01) between the Medimachine II and enzymatic procedure. (**H**) Statistical histogram of intracellular ROS presence (H_2_O_2_ content). Two-way ANOVA with a Bonferroni’s multiple comparison test revealed a significant difference (* *p* < 0.05, ** *p* < 0.01) between the Medimachine II and enzymatic procedure. Protocol significance: Lysosomes *.

**Figure 12 biomolecules-12-00701-f012:**
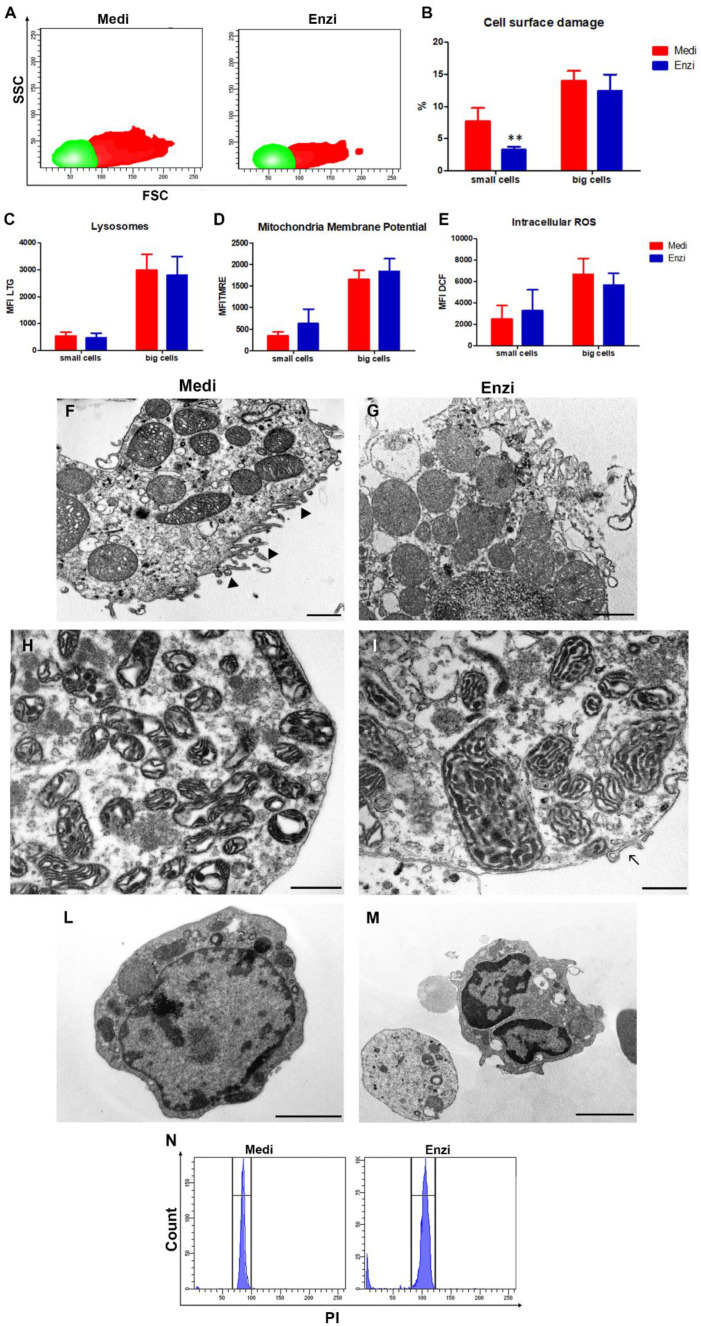
Study of fresh kidney. (**A**) Gating cell populations of small cells (green) and big cells (red) in the SSC versus FSC density plot for the further analysis. (**B**) Statistical histogram of cell surface damage percentage. Two-way ANOVA with a Bonferroni’s multiple comparison test revealed a significant difference (** *p* < 0.01) between the Medimachine II and enzymatic procedure. (**C**) Statistical histogram of lysosomal network. Two-way ANOVA with a Bonferroni’s multiple comparison test revealed a significant difference (** *p* < 0.01) between the Medimachine II and enzymatic procedure. (**D**) Statistical histogram of mitochondrial membrane potential (Δψ-MMP). Two-way ANOVA with a Bonferroni’s multiple comparison test revealed a significant difference (** *p* < 0.01) between the Medimachine II and enzymatic procedure. (**E**) Statistical histogram of intracellular ROS presence (H_2_O_2_ content). Two-way ANOVA with a Bonferroni’s multiple comparison test revealed a significant difference (** *p* < 0.01) between the Medimachine II and enzymatic procedure. (**F**,**G**) In both procedures, it is possible to distinguish cells from proximal convoluted tubules characterized by a relatively short brush border. Scale bar = 1 µm (**H**,**I**). Detail of the richness in the mitochondria of these cells and good conditions of preservation of cell and mitochondrial membranes. E, Scale bar = 1 µm; F, Scale bar = 500 nm. (**L**,**M**) TEM of leukocytes from the disaggregation procedures Scale bars = 2 µm. (**N**) Flow cytometry histograms of cell cycle analyses of both procedures.

**Figure 13 biomolecules-12-00701-f013:**
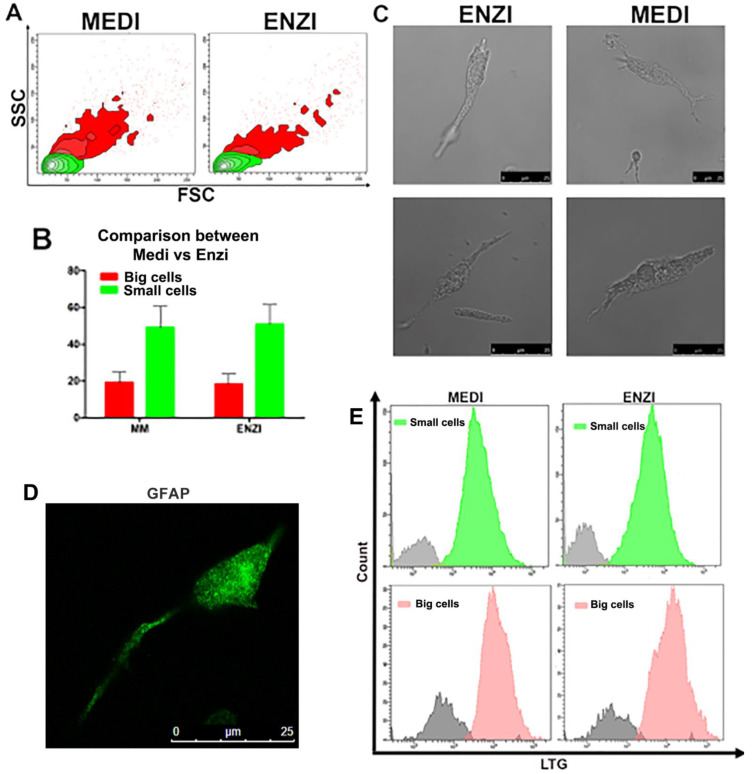
(**A**) Gating cell populations of small cells (green) and big cells (red) in the SSC versus FSC contour plot for the further analysis. (**B**) Flow cytometric data of percentages of cellular subpopulations. (**C**) Representative microscopic pictures (bright field) of cells obtained from both procedures. (**D**) Confocal images of GAFP-labeled cells. (**E**) Histogram of lysosomal compartment in big and small cells in both procedures.

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
