# Peer review of "Automated—Mechanical Procedure Compared to Gentle Enzymatic Tissue Dissociation in Cell Function Studies"

_biomolecules, 2022, doi:10.3390/biom12050701_

Round 1
Reviewer 1 Report
The authors present an approach for the digestion of different rat tissues through a mechanical procedure (Mediamachine II) in comparison to enzymatic digestion (carried out by treatment with trypsin). In addition to evaluating the number of viable cells extracted through the two different treatments, they assessed both from an ultrastructural point of view and the representation of mitochondria, lysosomes present, and relatively expression of ROS.
The choice of enzymatic treatment with trypsin (a purely digestive enzyme with the ability to recognize different sites for the substrate) and the exclusion of treatment with collagenase (pure) is undoubtedly limiting, even if understandable given the generic use of trypsin, in the processes of extracting cells from tissues and organs. Their exclusion of the use of collagenases was justified by referring to the work conducted by Crance et al. 2011 (doi: 10.1016 / j.jim.2011.07.002), which in their processes used extractive collagenases from C. Hystoliticum, containing various proteolytic enzymes, which has been widely described to have significant effects on both cell morphology and proliferative processes and differentiation. Today ultra-pure recombinant collagenases are used (with the addition of minimal concentrations of proteolytic enzymes) in a controlled digestive process, which preserve both proliferation and differentiation such as the viability of the extracted cells.
Therefore, it is suggested to remodel the introductory part by reporting the current knowledge about the extraction using collagenase and finalize the work, according to the comparison made, to obtain a sufficiently high number of functional cells in research applications.
Apart from these indications, a series of adjustments must be made into the article, namely:
pag 7 line 233: A-C, F, Bar = 1 µm (A,B are not morphology images); D,E, Bar = 2 µm (since not reported is supposed D, E-L = 2 µm);
pag 18 (Figure 9): are missing the identification letters from D to H;
pag 23 line 578: microvilli are identified by (*) fig 12 F - NO PRESENT THE MARKER;
pag 23 line 579: podocytes are identified by (°) fig 12 F - NO PRESENT THE MARKER.
Author Response
- We want to thank the reviewer for his/her/* suggestion. We have rewritten the introductory part, inserting several recent references, extending the concepts still present and, especially, reporting the current knowledge about the extraction using collagenase. Finally, both the abstract and the introduction, described a work finalized to obtain a sufficiently high number of functional cells, with specific cell functions emphasized by one or the other or both protocols (automated-mechanical and enzymatic) in research applications.
- We corrected what the reviewer suggested
- We added the identification letters
- The marker was substituted by the arrow, correctly unserted in the image
- The marker was substituted by the arrow, correctly unserted in the image

Reviewer 2 Report
I agree that cell isolation from tissues is incredibly important and the work done here is very in depth. The pictures of the intact sperm was exciting, but I would like more data to show that this was consistent. That said, the tests and the way the data was presented weakens my enthusiasm.
Major concerns.
- The introduction states that enzymatic treatment combined with mechanical stimulation maximizes the number of cells; however, this is not tested. Gentle mechanical stimulation combined with enzymatic would have been interesting. Different enzymatic treatments would have been interesting. The most gently method was chosen for comparison.
- The abstract states that the study points out how it is more ethical and scientifically sound to collect all tissues from one animal. This study does NOT look at this at all nor is this something that needs to be studied to understand. I do not understand this statement at all. Would a faster, less biased method help? Maybe but I am unconvinced based on the findings.
- I would have liked a better presentation of the data to compare between the tissue types. Even a mention in the conclusion would be helpful. It was hard to determine. I also am not sure I understand the benefit. In most cases it was similar to gentle enzymatic treatment and in some (ROS, for example) it was worse.
- The end of the introduction states that there was interest in assess long-term preservation of storing tissues outside of the body. This wasn't well addressed and seems outside the scope of cell isolation. This may be a wording issue.
- I can appreciate only looking at damage; however, I think that looking at the types of cells isolated is at least as interesting if not more. The small/large cells helped but was limited.
- The discussion is lacking. There is so much that could be said and may have influenced my enthusiasm but it feels like a lack of enthusiasm from the authors. Thoughts on how this might compare with other methods would be welcome. Discussion of limitations is also important.
Minor issues:
- A statistical analysis section would be welcome in the methods section. It is discussed in the figure legends and seems appropriate.
- There are too many one line paragraphs.
- Supplementary info - for the cell counts, how did you account for possible differences in starting material?
Author Response
- As suggested by Reviewer 1, the introductory part has been rewritten. Finally, both the abstract and the introduction, described a work finalized to obtain a sufficiently high number of functional cells, with specific cell functions emphasized by one or the other or both protocols (automated-mechanical and enzymatic) in research applications. As now stated, the goal of the work is not to indicate one best approach, but to provide information regarding the advantages and disadvantages of each protocol that we applied and their impact on the different managed tissues. We also stated that the choice of enzymatic treatment with trypsin (a purely digestive enzyme with the ability to recognize different sites for the substrate) and the exclusion of treatment with collagenase can appear limiting, however, given the generic use of trypsin in the processes of extracting cells from tissues and organs, we choose to apply this gentle enzymatic disaggregation to be compared with the automated-mechanical one, by MediMachine II (this point was also stated as a limitation of the work). We have applied the 2 procedures, but individually, to individuate the strengths and weaknesses of each. These results could be useful also to those researchers who are going to combine the 2 methods, although in our work we did not combine them but left them disjointed. This point is now stated also in the discussion.
- The sentence “Our study points out that it is more ethically and scientifically sound to collect all the possible information from the same sacrificed animal, analyzing all the organs and tissues” was deleted from the abstract, that was generally reformulated in several points. Although it was not appropriately inserted in the abstract nor appropriately explained, we want to try to explain it better now. What we wanted to say is that the possibility of being able to treat many organs and / or tissues with the same method, rapidly, could allow to perform for each animal, always, the maximum number of tests. This is sometimes hindered by the time needed or the numerous personnel that is required for many different procedures. However, this improper consideration, was excluded from the text and we thank the reviewer for suggesting deep pondering about the appropriateness of the sentence.
- We modified the description of several points of the RESULTS, in order to better present the comparison between the two methods applied on the different tissue types. In the DISCUSSION the points resuming the strengths and weaknesses of each disaggregation technique are now explained and further compared. As stated in the text of the manuscript, “the goal of the present study is not to indicate one best approach, but to provide information regarding the advantages and disadvantages of the protocols that we applied and their impact on the different managed tissues” Therefore, depending on the main scope of the research and on the types of tissue, it is preferable one method or the other, and, again, as the reviewer noted, for spleen disaggregation are both suitable. However, these considerations are now inserted in the text, as suggested by the reviewer.
- The introduction is now mostly reformulated and the scope of the evaluation of freezed/thawed cells should be now correctly addressed in our study plan. In fact, we show the functional response to freezing/thawing steps of the differently disaggregated cells. The titles of the paragraphs were accordingly modified.
- Comparisons are now better emphasized: the cell functions best highlighted by one procedure or the other are clearly mentioned in the discussion, The small/large differentiation, without a specific mAb labelling of the cells, is indicated as a limitation of the work. Unfortunately, some of the needed mAbs are commercially lacking and the cost of all would still be very high. These are the reasons why our future in depth analyses will be limited to spleen and liver only.
- We want to really thank the referee, for all suggestions, but for this last, particularly. Discussion was inserted and peculiarities of the study, together with its limitations are explained in the text. Apart from the corrections, or rather, the reformulation of the whole discussion, what we would like to transmit is the enthusiasm to which the reviewer referred
and that he was right to point out (and we are grateful for that). Limitations are now clearly mentioned in the discussion.
Minor issue:
- The statistical analysis section was added in the Methods section, as suggested by the referee
- The most one line paragraphs (mostly present in the Materials and Methods session) have been merged: now the text is less fragmented
- We processed with the 2 different methods tissue fragments of about 1 mm in diameter, as detailed in MM session. However, to minimize possible differences in starting material, we performed tests starting from 1 fragment only and from 3 or 4 fragments pooled together.

Round 2
Reviewer 2 Report
This is much improved, thank you.
Two minor items:
I'm still concerned with the overall cell yield. You mince into small sections; however, do you include all of the tissue from one organ or just a few sections. This could influence the results significantly. For future, normalizing to wet weight may be appropriate.
Some of the new text in the Results section is actually discussion. For some, this may be appropriate as there is so much information contained that the context may be lost by the discussion section; however, discussion of ROS overall would be appropriate in the discussion.
Author Response
We want to thank the reviewer that help us to improve the Manuscript. We enclose the responses to the last two items.
I'm still concerned with the overall cell yield. You mince into small sections; however, do you include all of the tissue from one organ or just a few sections. This could influence the results significantly. For future, normalizing to wet weight may be appropriate.
We used the same number of 1 mm3 pieces for the enzymatic (enzi) and for the automated mechanical (medi) disaggregation as following: first of all, 1 piece of 1mm3 then 4 pieces, obtained by the same organ. We will adopt for future the normalization to wet weight.
Some of the new text in the Results section is actually discussion. For some, this may be appropriate as there is so much information contained that the context may be lost by the discussion section; however, discussion of ROS overall would be appropriate in the discussion.
Following the suggestion of the reviewer, we moved some sentences detailing ROS functions to the DISCUSSION, as highlighted in the text.